# A HSV1 mutant leads to an attenuated phenotype and induces immunity with a protective effect

Xingli Xu[1], Xiao Feng[1], Lichun Wang[1], Ting Yi[2], Lichun Zheng[2], Guorun Jiang[1], Shengtao Fan[1], Yun Liao[1], Min Feng[1], Ying Zhang[1], Dandan Li[1], Qihan Li[1] *

**1** Institute of Medical Biology, Chinese Academy of Medical Sciences and Peking Union Medical College, Yunnan Key Laboratory of Vaccine Research and Development on Severe Infectious Diseases, Kunming, China, **2** Weirui Biotechnology (Kunming) Co., Ltd, Kunming, China

* liqihan@imbcams.com.cn

**Data Availability Statement:** All relevant data are within the manuscript and its Supporting Information files.

**Funding:** This work was supported by the National Natural Science Foundation of China (XL X, NO.

## Abstract

Herpes simplex virus type 1 (HSV1) is a complicated structural agent with a sophisticated transcription process and a high infection rate. A vaccine against HSV1 is urgently needed. As multiple viral-encoded proteins, including structural and nonstructural proteins, contribute to immune response stimulation, an attenuated or deficient HSV1 vaccine may be relatively reliable. Advances in genomic modification technologies provide reliable means of constructing various HSV vaccine candidates. Based on our previous work, an M6 mutant with mutations in the *UL7*, *UL41*, *LAT*, *Us3*, *Us11* and *Us12* genes was established. The mutant exhibited low proliferation in cells and an attenuated phenotype in an animal model. Furthermore, in mice and rhesus monkeys, the mutant can induce remarkable serum neutralizing antibody titers and T cell activation and protect against HSV1 challenge by impeding viral replication, dissemination and pathogenesis.

## Author summary

Herpes simplex virus type 1 (HSV1) may lead to various herpes infections and can establish latent infection in neurons, which shows a high infectivity rate in populations. There is currently no effective therapeutic and prophylactic vaccine against HSV1. Previous studies have indicated that attenuated or deficient HSV1 vaccines are capable of safely eliciting a comprehensive protective immune response. With the advances in genomic modification technologies and accumulated knowledge of viral-encoded proteins, various predictable vaccine candidates can be reliably designed. In this work, a mutant M6 containing mutations in the *UL7*, *UL41*, *LAT*, *Us3*, *Us11* and *Us12* genes was established. Furthermore, the proliferation characteristics, virulence phenotype and immunogenicity of the mutant were analyzed in different cells and in mice and rhesus macaques.

81802868 and QH L, NO. 31670173), Fundamental Research Funds for the Central Universities (XL X, NO. 3332018129), the CAMS Initiative for Innovative Medicine (QH L, NO. 2016-I2M-1-019) and the Science and Technology Major Project of Yunnan Province (MF, NO. 2017ZF006 and YZ, NO. 2017ZF020). The funders had no role in the study design, data collection and analysis, the decision to publish, or the preparation of the manuscript.

**Competing interests:** The authors have declared that no competing interests exist.

## Introduction

Herpes simplex virus type 1 (HSV1), a double-stranded DNA virus with a complicated genomic structure and transcriptional mechanism [1, 2], is an agent that leads to various herpes infections, more commonly causing oral vesicles, corneal herpes and approximately 47% of first-time genital herpes cases [3, 4]. Analysis of the etiology and sero-epidemiology of HSV1 indicates infection rates of up to 20–60% for this virus in populations of various ages worldwide [5–8]. The characterized pathological process of local primary infection in a tissue, such as the oral mucosa, followed by latent infection of neurons and reactivation under some circumstances, is related to various unidentified interactions between viral molecules and the host [9, 10]. HSV1 is a great challenge in regard to clinical treatment and preventive vaccination. To date, much of the data provided by studies of HSV vaccines focusing on the safety and validity of immune responses suggest that much effort still needs to be exerted to elicit clinical protective efficacy by immunization with an HSV vaccine [11, 12]. Recent reports on deficient HSV vaccines suggest that multiple proteins encoded during viral infection, including structural and nonstructural molecules, contribute to the comprehensive antigenic signals required for a specific immune response [13–15]. In this case, it is reasonable to develop an attenuated or deficient HSV1 vaccine because these vaccines are capable of eliciting a comprehensive protective immune response with the premise of ensured safety [16–19]. Advances in genomic modification technologies provide reliable means of reconstructing HSV genomic components and enable the design of various predictable vaccine candidates for analysis of their immunogenicity [20, 21]. The interactions of viral-encoded proteins and the immune system may be revealed via the identification of characterized properties of various viral mutants [19, 22, 23]. The data obtained from these efforts will be significant in guiding the technical strategy for HSV vaccine development. In our previous studies, a series of HSV1 mutants with deficiencies in different genes, including the *UL7* gene involved in viral transcriptional regulation, the *UL41* gene related to virulence and the LAT gene involved in latency establishment, were constructed and identified to present different levels of attenuation [24, 25]. Based on these works, we constructed a new HSV1 mutant strain with mutations in the *UL7*, *UL41*, *LAT*, *Us3*, *Us11* and *Us12* genes, which are responsible for viral genomic transcription, virulence, latent infection, host cellular apoptosis inhibition and antigen presentation [10, 24, 26–30]. The biological characteristics and attenuated phenotype of this mutant, especially its potential to elicit immunoprotective effects, were analyzed and evaluated in cells and animal models in this work.

## Results

### Construction of an HSV1 strain with the assembly of mutations in 6 genes and identification of its biological characteristics

Our previously reported data indicated that compared with the WT strain, the HSV1 strain M3 with mutated *UL7*, *UL41* and *LAT* genes presented a typical attenuated phenotype in infected mice [25]. Based on this M3 strain, mutations in the *Us3*, *Us11* and *Us12* genes were made via CRISPR/cas 9 in which the fragments from 1248 to 1393, 443 to 649 and 1044 to 1250 in the nucleic acid sequences of *Us3*, *Us11* and *Us12*, respectively, were deleted with different primers (Fig 1A). This mutant was named M6.The M6 strain was cloned and identified by specific primers (Fig 1B; S1 and S2 Texts),and its mutated double copies of *Us3*, *Us11* and *Us12* genes were confirmed through sequencing of the PCR-amplified genomic fragments using specific primers (S1 Fig; S1 Table). Further detection of its biological properties suggested very low proliferative kinetics presenting as very small plaques (Fig 1C) and slow growth curves in epithelial cells, including human embryonic fibroblasts (KMB17) (Fig 1D)

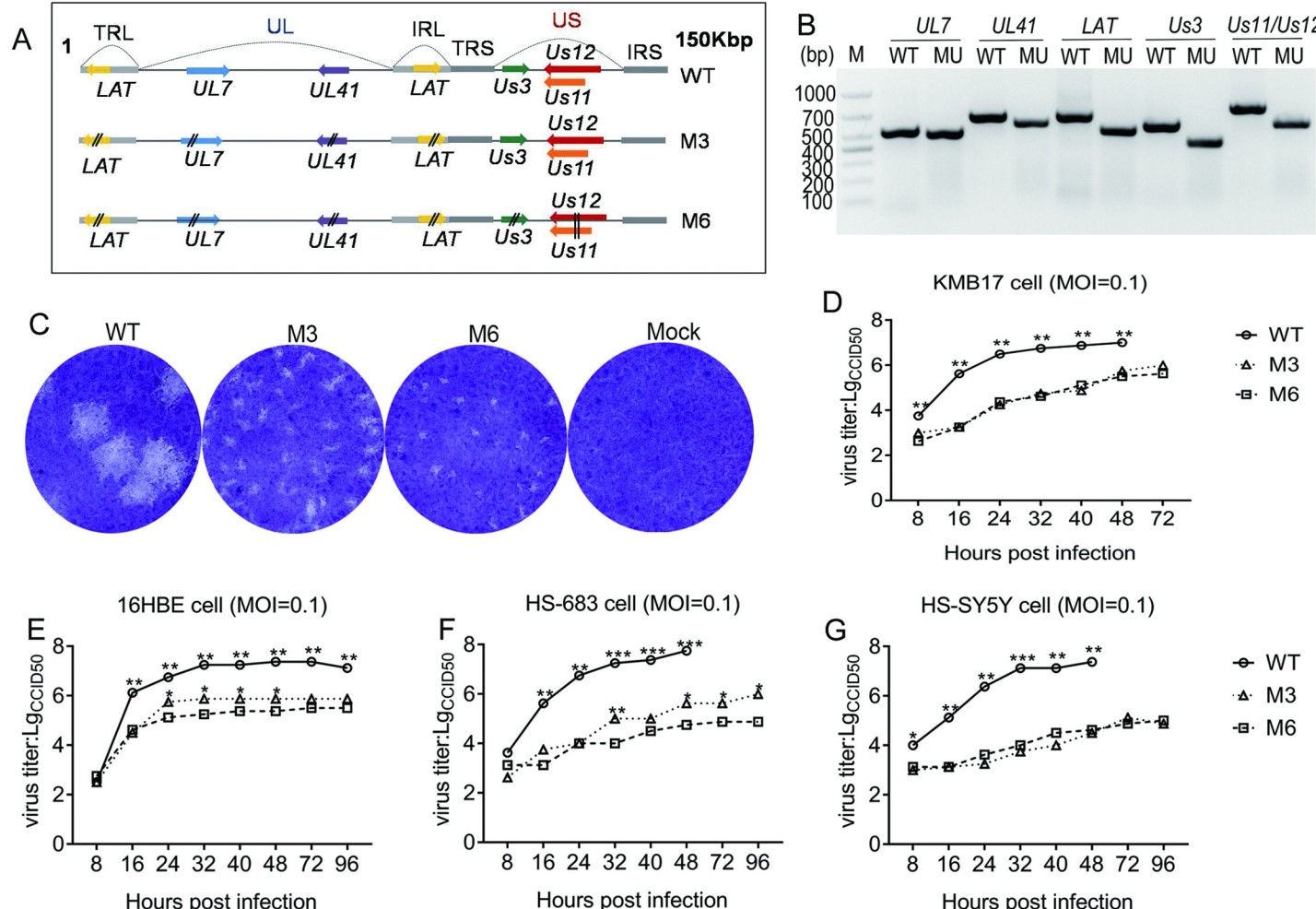

**Fig 1. Preliminary analysis of the HSV1 mutant M6 in cultured cells.** (A)Schematic of the gene modifications in the M6 mutant. (B) Electrophoresis results for the mutated *Us11* and *Us12* genes in the M6 mutant. The mutation was identified via PCR. (C) The plaque morphology of the M6 mutant in KMB17 cells. Growth curves for the WT, M3 and M6 strains grown in KMB17 cells (D), 16HBE cells (E), HS-683 cells (F) and HS-SY5Y cells (G).

and bronchial epithelial cells (16HBE) (Fig 1E), for the M6 strain compared with the primary M3 strain and WT strain (McKrae strain). The M6 strain also showed similarly low proliferative trends in neurogliocytoma cells (Hs683) (Fig 1F) and neuroblastoma cells (SH-SY5Y) (Fig 1G). These results indicated that the assembly of mutations in the *Us3*, *Us11*and *Us12* genes, which enable the inhibition of cellular apoptosis and cellular protein synthesis and block antigen presentation during viral infection [28, 29, 31, 32], is capable of reducing viral proliferative and invasive capacities in cells.

## Attenuated phenotype of the M6 strain compared with that of the WT strain in infected mice

Our previous report indicated the attenuated phenotype of M3 with mutated *UL7*, *UL41* and *LAT* genes in a mouse model [25]. In the observation of M6 infection of Balb/c mice, we obtained similar results that indicated a significantly attenuated phenotype for the M6 strain compared with the WT strain. We observed asymptomatic animals inoculated with M3 and M6 compared with serious manifestations including ulcerative lesions in the eyes in control

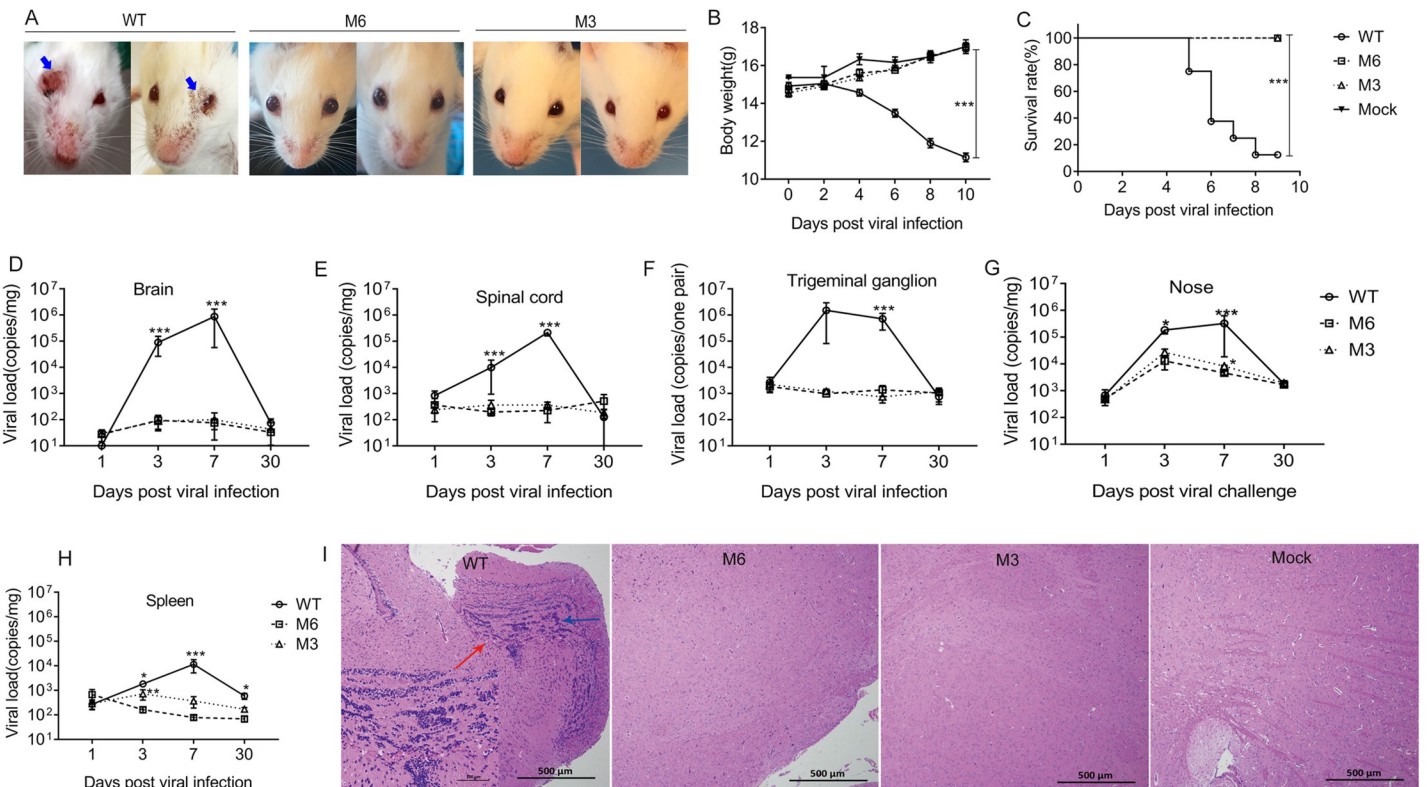

**Fig 2. The attenuated phenotype of the M6 strain in mice. (A)** Manifestations including ulcerative lesions in the eyes (blue arrow) of infected mice. **(B)** Weights of mice infected with the WT strain (circle),M6 strain (square) or M3 strain (triangle) or mock infected (PBS, inverted triangle) over a 10-day observation period.**(C)** Survival curves for the infected mice during the 10-day observation period. Viral loads in the brain **(D)**, spinal cord **(E)**, trigeminal ganglion **(F)**, nose **(G)** and spleen **(H)** of mice infected with the WT, M6 or M3 strain, as determined by RT-qPCR. The data are shown as the mean ± SEM based on data from three independent mice. *$p < 0.05$; **$p < 0.01$;***$p < 0.001$. **(I)** Pathological changes in the cerebrum of mice inoculated with the WT strain, M6 strain, M3 strain or PBS. Tissue sections were stained with hematoxylin and eosin and imaged. Tissue hyperemia and inflammatory cell infiltration detected at 7 d.p.i. are indicated with red and blue arrows, respectively. Scale bars = 500 μm or 100 μm.

animals inoculated with the WT strain (Fig 2A). There were significant differences in body weight and survival rate among these groups (Fig 2B and 2C). The detection of the viral loads in various organs of mice sacrificed at the specific time point in the three groups suggested that the mice infected with the WT strain presented a dynamic viral proliferation curve (Fig 2D–2F), especially in the brain, trigeminal ganglia and spinal cord tissues, which were previously confirmed to be specific targets for viral infection [25, 33, 34]; however, less viral replication was found in most tissues, including the trigeminal tissues, in the M3- and M6-infected mice, except a slight growth in nasal and spleen tissues (Fig 2G and 2H). Furthermore, the viral loads in the nasal and spleen tissues of the M6-infected mice were significantly lower than those of the M3-infected mice. These data suggest that M6 proliferation might slightly occur in local epithelial tissues and immune cells, whereas proliferation of the WT strain occurs in various tissues. Further histopathological observation indicated that only slight infiltration of inflammatory cells into the cerebrum tissues were found in mice inoculated with M6 (Fig 2I), whereas obvious inflammatory lesions were identified in the tissues of mice infected with the WT strain (Fig 2I). All of these data suggest that the attenuated phenotype of M6 enables its safety, which is a requirement for attenuated vaccine candidates mandated by the World Health Organization (WHO).

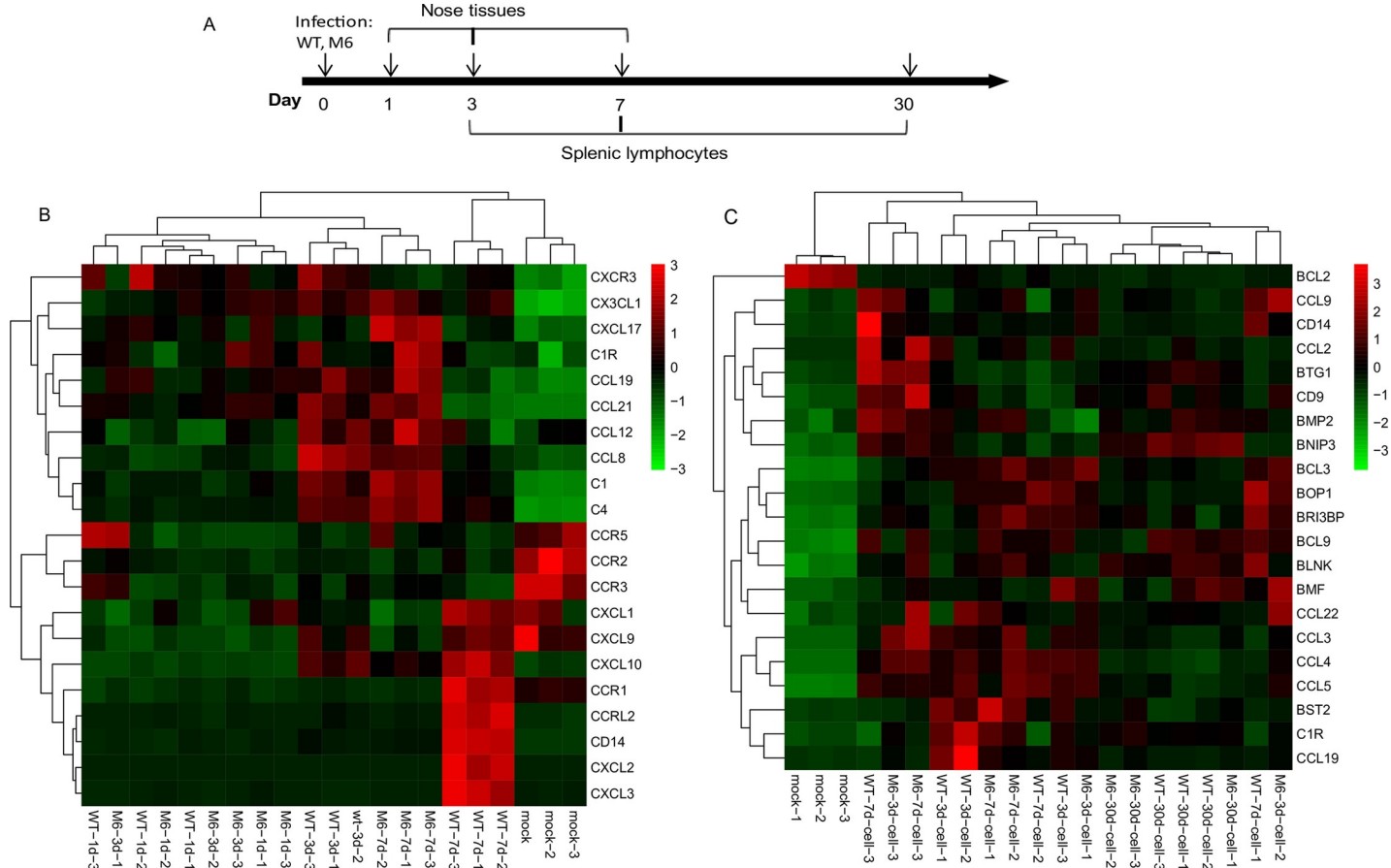

**Fig 3. The immune responses induced by WT or M6 virus in mice by transcript omics analysis. (A)** Flow charts for sample collection for transcript omics analysis. Genes with significantly upregulated or downregulated expression in nasal tissues **(B)** or splenic lymphocytes **(C)** and linked to IFN-γ by INTERFEROME analysis.

### The mRNA profile in mice inoculated with M6 compared with that in WT strain-infected mice suggests a characteristic immune response initiated by M6

Usually, the study of HSV vaccines addresses protection against viral challenge, and specific neutralizing antibody and cytotoxic T lymphocyte (CTL) responses are recognized as important immunological indicators [17, 35]. However, to understand the comprehensive immune response producing a protective effect and evaluate immunological indicators, omics analysis of mRNA expression is valuable [36]. Although the mechanism by which HSV induces an immune response is not clear [34], our work using a comparison between characterized immunity and the mRNA expression profiles of the locally infected tissue (nose) and splenic lymphocytes of mice inoculated with M6 and the WT strain investigated the immune response required for vaccine effectiveness (Fig 3A). The characterized mRNA expression in mice inoculated with M6 compared with that in mice infected with the WT strain indicated that in the local tissue genes related to innate immunity were enriched in the complement pathway, antigen presentation, the PI3K-Akt signaling pathway and IgA production (Table 1). Compared with the WT strain, M6 also elicited dynamic expression of mRNAs related to the IFN response (Table 1). Furthermore, the omics analysis of the mRNA gene profiles of splenic lymphocytes suggested that upregulated expression of genes related to antigen presentation

**Table 1. The enriched KEGG pathways of nasal tissues associated with the immune response with a *p*-value <0.05 and *p*-value in the lowest 20 at each time point.**

| pathways (WT *vs* M6) | 1d.p.i. (*p*-value) | | 3d.p.i. (*p*-value) | | 7d.p.i. (*p*-value) | |
|---|---|---|---|---|---|---|
| | WT upregulated | M6 upregulated | WT upregulated | M6 upregulated | WT upregulated | M6 upregulated |
| Herpes simplex infection | / | 0.026 | 4.716e-16 | / | | 9.101e-05 |
| Cytosolic DNA-sensing pathway | / | / | 3.732e-06 | / | | |
| RIG-I-like receptor signaling pathway | / | / | 0.0001 | / | | |
| Antigen processing and presentation | / | / | 0.0002 | / | | |
| Natural killer cell-mediated cytotoxicity | / | / | 0.0003 | / | | |
| Toll-like receptor signaling pathway | / | / | 0.0006 | / | 8.359e-06 | |
| Chemokine signaling pathway | / | / | 0.002 | / | | |
| Cytokine-cytokine receptor interaction | / | / | / | / | 6.758e-12 | |
| TNF signaling pathway | / | / | / | / | 1.919e-06 | |
| NF-kappa B signaling pathway | / | / | / | / | 3.371e-05 | |
| Jak-STAT signaling pathway | / | / | / | / | 3.858e-05 | |
| Intestinal immune network for IgA production | / | / | / | / | | 6.178e-06 |
| PI3K-Akt signaling pathway | / | / | / | / | | 0.0070 |
| Antigen processing and presentation | / | / | / | / | | 0.0070 |

occurred in the mice inoculated with M6, whereas the upregulation of genes involved in the inflammatory reaction occurred in mice inoculated with the WT strain (Table 2). The data were further confirmed through dozens of genes' quantitative RT-PCR (qRT-PCR) by the ΔCt value (S2 and S3 Figs; S2 Table). Similarly, different dynamic trends in gene responses related to the IFN family were observed in both groups (Fig 3B and 3C). These results suggested that the assembly of mutations in 6 genes led to specific viral antigenic stimulation of the immune system, followed by a characterized systemic immune response. Thus, the immunological phenotype, especially the immunoprotective efficacy in mice elicited with M6, should be further investigated.

**Table 2. The enriched KEGG pathways of splenic lymphocytes associated with the immune response with a *p*-value <0.05 and *p*-value in the lowest 20 at each time point.**

| pathways (WT *vs* M6) | 3d.p.i (*p*-value) | | 7d.p.i.(*p*-value) | | 30d.p.i. (*p*-value) | |
|---|---|---|---|---|---|---|
| | WT upregulated | M6 upregulated | WT upregulated | M6 upregulated | WT upregulated | M6 upregulated |
| Herpes simplex infection | 0.0006 | / | / | / | / | 0.017 |
| Cytosolic DNA-sensing pathway | 0.0013 | / | / | 0.024 | / | 0.012 |
| RIG-I-like receptor signaling pathway | 0.0016 | / | / | 0.027 | / | 0.015 |
| Antigen processing and presentation | / | / | / | 0.00047 | / | / |
| Natural killer cell-mediated cytotoxicity | / | / | 2.147e-05 | / | / | / |
| Toll-like receptor signaling pathway | / | / | / | 0.039 | / | / |
| TNF signaling pathway | / | / | / | / | 0.038 | / |
| NF-kappa B signaling pathway | / | / | 1.501e-05 | / | / | / |
| Intestinal immune network for IgA production | / | / | 5.976e-09 | / | / | / |
| PI3K-Akt signaling pathway | / | / | 0.0004 | / | / | / |
| B cell receptor signaling pathway | / | / | 4.189e-06 | / | / | / |
| Fc epsilon RI signaling pathway | / | / | 3.150e-06 | / | / | / |
| Fc gamma R-mediated phagocytosis | / | / | 6.906e-06 | / | / | / |
| T cell receptor signaling pathway | / | / | / | / | 0.012 | / |

## Immunity to M6, an experimental attenuated vaccine, produces an enhanced immune response and effective protection against viral infection in mice

In the experimental design of this work, we raised the hypothesis that a mutation of some virally encoded proteins with functions inhibiting different key molecules in the innate and adaptive immune responses could lead to the host immune system developing an integrated response against viral antigens. Based on this hypothesis, we first immunized mice with three doses of M6 via the intramuscular route followed by T cell proliferation and neutralizing antibody detection at 7, 14 and 28 d.p.i. In the observation of T cell responses, the increased numbers of CD4+ and CD8+ T cells producing IFN-γ were detected in M6-immunized mice with different doses (Fig 4A and 4B). Additionally, neutralizing antibody titers were increased to 1:4 (GMT) at 28 d.p.i in the M6-immunized mice (Fig 4C). Combining the T cell responses and neutralizing antibody titers, the medium dose (1x10$^4$) was inferred here to be the suitable dose to induce an adaptive immune response. Then, more details of the specific immune response elicited by M6 inoculation via the intramuscular route were investigated in mice after the safety of M6 was identified as described above (Fig 4D). However, given the relatively low titers of neutralizing antibodies in the animals at 25 d.p.i., the mice were boosted with M6. Further detection at 25 days after the boost showed 100% seroconversion and obviously increased antibody titers (Fig 4E); meanwhile, increased T cell proliferation against IFN-γ was revealed by antigen stimulation with a viral membrane protein (Fig 4F). Based on these data, we performed a challenge study using the WT strain (McKrae strain) to investigate the immunoprotective efficacy established by M6 at 30 days post-boost. The study indicated that no clinical symptoms were observed in the M6 group (M6-muscle) compared with the positive control group (Mock-Mck) after WT virus challenge (Table 3). The detection of viral load in the peripheral blood did not show a kinetic curve indicative of viremia in either group (Fig 4G). Importantly, the proliferation of the challenge virus in nervous tissues, including the brain, spinal cords and trigeminal ganglia, was restricted in the boosted group, whereas the positive control group exhibited typical viral replication in tissues (Fig 4H–4J). The histopathological analysis of various organs indicated that no pathological changes were observed in the mice in the immunized group, except for slight infiltration by a few inflammatory cells, compared with those in the positive control group (Fig 4K). These results suggest that M6 elicits effective immunoprotection, as it induces specific neutralizing antibody and CTL responses, and confirm the significance of omics analysis of mRNA profiling of the response in M6-inoculated mice, as described above.

## M6 shows effective immunity in rhesus macaques with elicitation of a neutralizing antibody response and protection against virus attack

Previous studies on HSV have recognized that mice are an effective and advantageous animal model and enable the development of useful immunological data for vaccine development [17]. However, due to the limitations related to the distant genetic relationship between mice and humans, it is difficult to reflect the true immunoprotective process in humans [37]. In our previous work, the infectious process and associated pathological characteristics in rhesus macaques infected with HSV1 were investigated and described, and this animal was identified as susceptible to HSV1 infection with clinical pathological processes consisting of oral vesicles, typical viremia, dynamic viral proliferation in various tissues including nervous tissue and latent virus reactivation in the trigeminal tissue under some circumstances [38]. Here, the immunogenicity of M6 was evaluated in rhesus macaques via detection of their neutralizing antibody response and T cell response in a dose-dependent immunization test via the

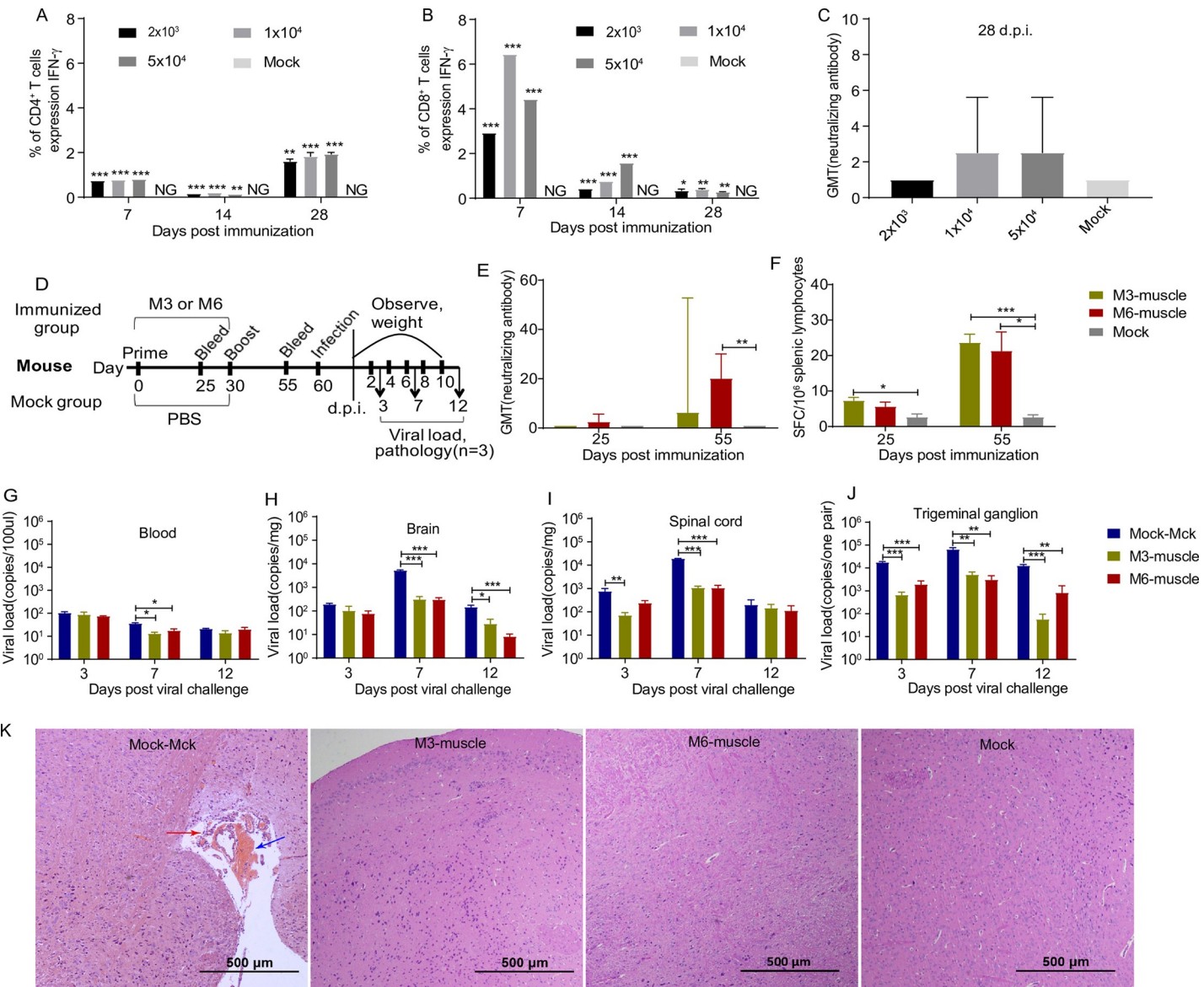

**Fig 4. The immunoprotective efficiency of the strain M3 and M6 in a mouse model. (A, B)** T cell responses to three doses of M6 immunization. Peripheral blood lymphocytes were isolated from immunized mice at 7,14 and 28 d.p.i. and stimulated with PMA, ionomycin and Golgi stop, followed by intracellular staining for IFN-γ. **(A)** Percentages of CD4+ T cells that were producing IFN-γ. **(B)** Percentages of CD8+ T cells that were producing IFN-γ. **(C)** Neutralizing capability of the antibodies against HSV1 in mice immunized with HSV1 M6 (three doses) or PBS. The geometric mean titers (GMTs) of the neutralization antibodies were measured by a neutralization test as described in "Materials and Methods." The GMT values for the negative control (PBS) groups were all <2. **(D)** Design of the mouse experiment. **(E)** The antibody titers specific for HSV-1 in mice infected with HSV-1 M3 or M6 or mock infected (PBS) (n = 3 for each) at 25 or 55 days after viral immunization. **(F)** ELISpot analysis of the IFN-γ-secreting cells within the splenic lymphocyte population in HSV-1 M3-, M6- or mock-infected mice. Splenic lymphocytes were incubated for 30 h in the presence of a stimulus. The positive control was phytohemagglutinin (PHA). **(G-J)** M3-, M6- or mock-infected mice were challenged with the WT virus at 30 days after viral infection. Viral loads in the blood **(G)**, brain **(H)**, spinal cord **(I)** and trigeminal ganglion **(J)** determined by RT-qPCR. The values are presented as the mean ± SEM. *$p < 0.05$; **$p < 0.01$; ***$p < 0.001$. **(K)** Pathological changes in the cerebrum of virus-challenged mice. Tissue sections were stained with hematoxylin and eosin and imaged. Tissue hyperemia and inflammatory cell infiltration detected at 7 d.p.i. are indicated with red and blue arrows, respectively. Scale bars = 500 μm or 100 μm.

intramuscular route (Fig 5A).The results showed that immunizing monkey with three doses can induce significant proliferation of CD4+ and CD8+ T cells with IFN-γ specificity (Fig 5B and 5C).However, the neutralizing antibody titers showed a low titer until 28 d.p.i. (Fig 5D). Then, the immunogenicity and protective effect of M6 were further investigated in a challenge

**Table 3. Manifestations of M3- or M6-immunized or mock mice infected with wild-type virus (%) (N = 12).**

| Symptoms | Mock-Mck | M3-muscle | M6-muscle |
|---|---|---|---|
| Back arching | 87.5 | 0 | 0 |
| Inverted hair | 100 | 0 | 0 |
| Blindness | 62.5 | 0 | 0 |
| Paralysis | 20 | 0 | 0 |
| Death | 83.3 | 0 | 0 |

test with the medium dose (Fig 6A).In this assay, the monkeys were boosted with M6 at 60 d.p.i. The results suggested seroconversion in all macaques inoculated twice with M6 via the intramuscular route (Fig 6B). Then, these immunized animals were challenged with WT strain. The result showed these animals were asymptomatic, while all of the animals in the positive control group developed oral vesicles or exhibited redness around the eyes after WT strain challenge (Fig 6C). The detection of the viral loads in the peripheral blood and oral and eye excreta at 12 days post-challenge suggested dramatically lower viral loads in the M6-inoculated group than in the positive control group (Fig 6D–6F). Further detection of the viral loads in

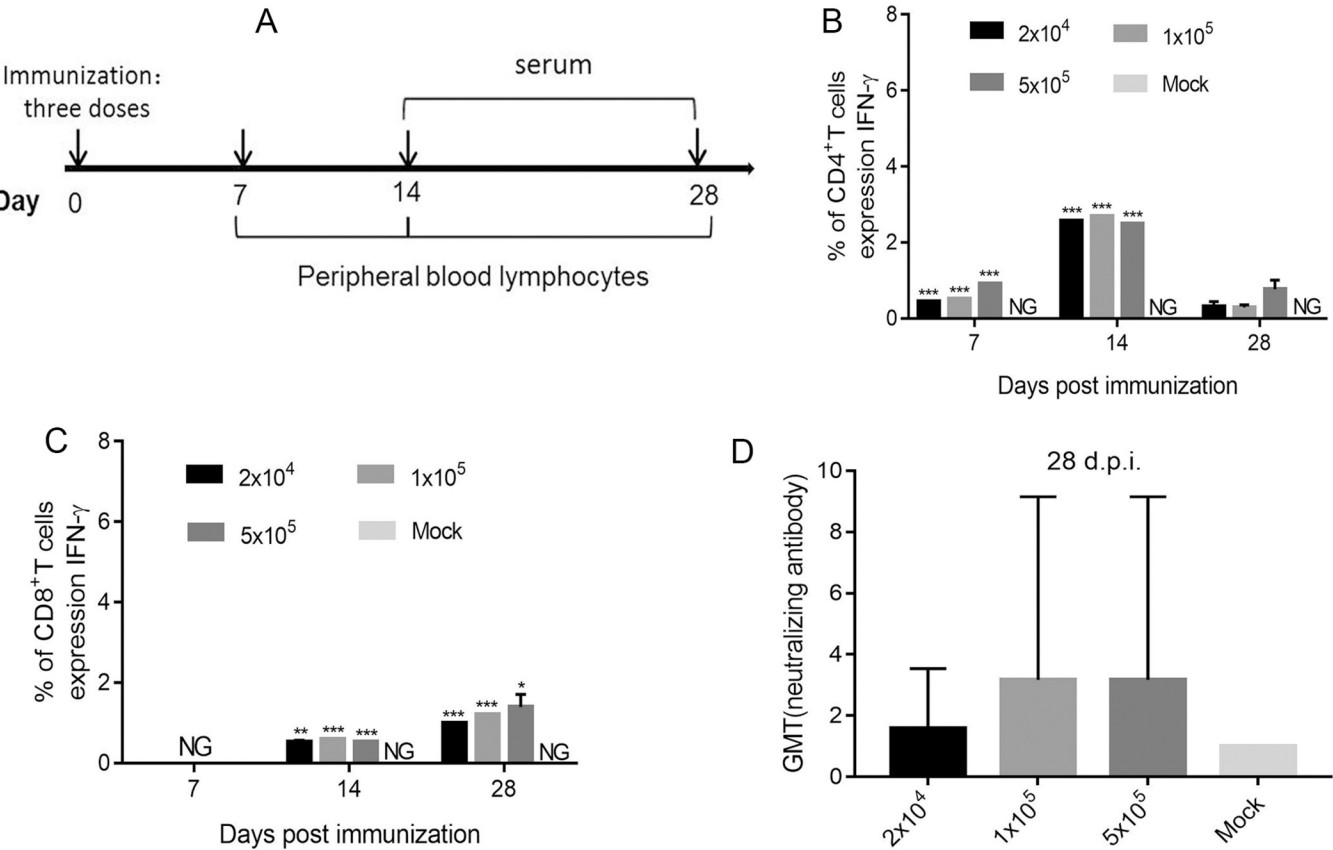

**Fig 5. The rhesus monkeys immunized with mutant M6 could induce significant adaptive immune response. (A)** Design of the monkey dose-dependent experiment. **(B, C)** T cell responses to three doses of M6 immunization. Peripheral blood lymphocytes were isolated from immunized monkeys at 7,14 and 28 d.p.i. and stimulated with PMA, ionomycin and Golgi stop, followed by intracellular staining for IFN-γ. **(B)** Percentages of CD4[+] T cells that were producing IFN-γ. **(C)** Percentages of CD8[+] T cells that were producing IFN-γ. **(D)** Neutralizing capability of the antibodies against HSV1 in mice immunized with HSV1 M6 (three doses) or PBS. The geometric mean titers (GMTs) of the neutralization antibodies were measured by a neutralization test as described in "Materials and Methods." The GMT values for the negative control (PBS) groups were all <2.

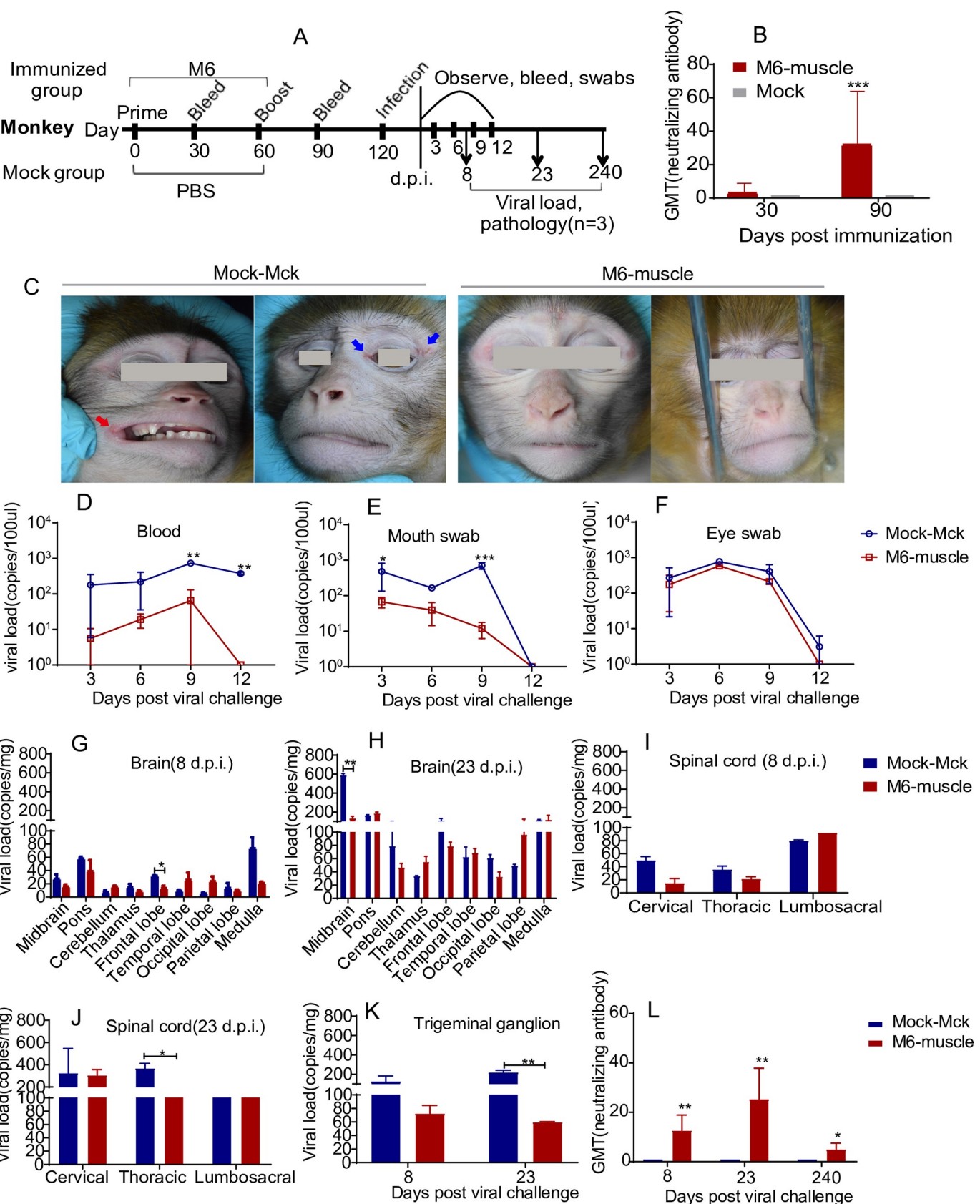

**Fig 6. The immunoprotective efficiency of the strain M6 in a monkey model. (A)** Design of the monkey experiment. **(B)** The neutralizing capability of antibodies specific for HSV-1 from M6-infected or mock-infected monkeys. **(C)** Manifestations including ulcerative lesions in the mouth (red arrow) or eye (blue arrow) of infected monkeys.**(D-F)**Viral loads in the blood**(D)**, mouth swabs **(E)** and eye swabs **(F)** of infected monkeys during 12 days after viral infection. **(G-K)** Viral loads in the brain **(G, H)**, spinal cord **(I, J)** and trigeminal ganglion **(K)** of infected monkeys at 8 or 23 d.p.i. **(L)**The neutralizing capability of antibodies specific for HSV-1 fromM6-infected or mock-infected monkeys at 8 or 23 d.p.i. or 8 month post-immunization. The values are presented as the mean ± SEM. $^*p< 0.05$; $^{**}p< 0.01$; $^{***}p< 0.001$.

various organs of the animals sacrificed at day 8 or 23 post-challenge revealed reduced and nondynamic proliferation in the animals in the M6-inoculated group, whereas dynamic replication occurred in the macaques in the positive control group (Fig 6G–6K).Histopathological detection indicated that viral challenge elicited slight aggregation of some inflammatory cells in the lungs and spinal cord in the M6-inoculated group (Table 4) but induced some inflammatory reactions in the brain and spinal cord and vesicle lesions in the oral mucosa in the

**Table 4. Pathological analysis results of M6-immunized or mock monkey infected with wild-type virus (%) (N = 3).**

| organs | | 8 d.p.i. | | 23 d.p.i. | | 240 d.p.i. | |
|---|---|---|---|---|---|---|---|
| | | Mock-Mck | M6-muscle | Mock-Mck | M6-muscle | Mock-Mck | M6-muscle |
| Nervous system | Cerebrum | + | - | + | - | - | - |
| | Cerebellum | - | - | + | -/+ | -/+ | - |
| | Spinal cord | + | + | ++ | -/+ | -/+ | - |
| | Trigeminal nerve | + | + | + | -/+ | -/+ | - |
| | Dorsal root nerve | + | -/+ | - | - | - | - |
| | Other nerves | - | - | - | - | - | - |
| Immune system | Palatine tonsil | - | - | - | - | - | - |
| | Tongue tonsil | - | - | ++ | -/+ | - | - |
| | Pharyngeal tonsil | + | + | -/+ | -/+ | - | - |
| | Axillary lymph node | + | + | | - | - | - |
| | Other lymph node | + | - | | - | - | - |
| Respiratory system | Lung | + | - | + | - | -/+ | - |
| | Trachea | + | - | + | - | -/+ | - |
| | Bronchus | + | - | + | + | -/+ | - |
| Digestive system | Esophagus | + | + | - | - | - | - |
| | Colon | + | + | - | - | - | - |
| | Other intestines | | - | - | - | - | - |
| Other | Nose | -/+ | -/+ | - | - | - | - |
| | Eyelid | + | -/+ | + | -/+ | - | - |
| | Lips | + | -/+ | - | - | - | - |
| | Bladder | + | -/+ | - | - | - | - |
| | Other organs | - | - | - | - | - | - |

Other nerves: oculomotor nerve, optic nerve, sciatic nerve and intercostal nerve

Other lymph nodes: Submandibular gland lymph node, pulmonary lymph nodes, intestinal lymph nodes and abdominal lymph nodes

Other intestines: duodenum, jejunum, ileum and rectum

Other organs: heart, liver, kidney, spleen, pancreas, testis and muscle

Scale as follows

Grade -, normal tissue.

Grade+, slight infiltration of inflammatory cells without neural damage.

Grade++, slight damage with inflammatory cell infiltration.

Grade+++, massive damage with inflammatory cell infiltration.

Grade++++, serious damage with or without inflammatory cell infiltration.

positive control animals (Table 4), whereas no viral load was identified in both M6- and WT-infected group at 8 months post-challenge. Pathologic observation of the animals at 8 months post-challenge also indicated that no inflammatory cells were observed in the M6-inoculated group, while slight inflammatory cells did exist in positive control animals (Table 4). Finally, the animals in the M6-inoculated group showed increased neutralizing antibody levels after viral challenge (Fig 6L). All of these data obtained from the macaque study further confirm that M6 can induce effective protection in primates against WT strain attack.

## Discussion

Studies on the pathogenesis of HSV reveal a complicated mechanism of viral components interacting with the human host, which leads to pathological interference in the innate immune response and adaptive immune response through virally encoded molecules altering or blocking antigen recognition, signal transduction by pathways and T cell activation *in vivo* [39, 40]. These events lead to a high rate of clinical cases of latent infection, clinical recurrence and re-infection [41–44] and make the development of an HSV1 vaccine difficult. Previous studies of HSV vaccines have focused on viral membrane proteins, such as gB, gC and gD [45–49]. With an improved understanding of the clinical pathogenesis of HSV1, the accumulated data on the roles that many viral components play in infection suggest that an integrated concept of a viral antigen system comprising not only membrane proteins but also capsid proteins and others encoded during viral infection should be a concern in vaccine development [50, 51]. The clinical trials assessing several experimentally deficient and attenuated HSV vaccines seem to highlight the trend of next-generation HSV vaccines. In this case, our hypothesis is that mutations in some viral genes responsible for interfering with the host response provide a clear circumstance for establishing an effective immune response via modulation of viral infective characteristics, viral antigen presentation and immune activation. Based on our previous work, *UL7* and *UL41* gene mutations were used as measures to control viral proliferation and modulate virulence [24, 25], *Us3* and *Us11* gene mutations were used to decrease viral interference with the cellular stress response [52, 53], *Us12* gene mutation was capable of maintaining normal antigen presentation [29], and *LAT* gene mutation was expected to restrict latent viral infection [27].

The observations from the study of M6 in cells and animals indicate that the assembly of 6 genes creates a characteristic attenuated phenotype that shows small plaques with diameters smaller than 1/10 of those induced by the WT strain and very slow proliferative rates in various cells, including epithelial cells and neurons. Importantly, mice inoculated with this strain do not show any clinical manifestations or histopathological lesions and present with very low levels of viral proliferation in various organs, especially in nervous tissues such as the trigeminal ganglions. These data support the safety of M6 as an experimental vaccine candidate. Furthermore, the low level of viral proliferation found in local epithelial tissue and the spleen consisting of various immune cells implied a potential advantage for M6 as a vaccine candidate that could be recognized by the innate immune system and then by T cells in epithelial and immune tissues. The omics analysis of the mRNA profiles of the local tissue and splenic lymphocytes for the M6 group compared with the same profiles for the WT group supports this prediction by demonstrating sequential upregulation of the expression of genes related to the complement pathway, antigen presentation and the NF-kB transcriptional system, especially the dynamic expression of genes in the IFN family. The integrated immune response is represented by specific neutralizing antibodies and responsive T cell proliferation in mice or rhesus macaques inoculated with M6 via the intramuscular route. The data showing similar trends in both types of animals may be referenced for further investigation of M6 in humans.

Significantly, the mice and macaques inoculated with M6 twice exhibited effective protection against WT strain challenge in which asymptomatic challenged animals were identified and very low viral proliferation was observed in various organs, especially in the trigeminal ganglion; this effect could be maintained in rhesus macaques at least for 8 months after viral challenge. Thus, the M6 strain exhibits the immunogenicity and safety needed in an attenuated vaccine that enables to prevent HSV1 disease in mice and rhesus macaques.

However, further studies on this strain are needed; constant immunological observation of inoculated animals, including mice and macaques, should be considered, and the potential for genomic recombination between M6 and the WT challenge strain *in vivo* should be evaluated at certain times.

## Materials and methods

### Ethics statement

The authors declare that they have no conflict of interest. All institutional and national guidelines for the care and use of laboratory animals were followed. The experimental protocols of mice were approved by the Yunnan Provincial Experimental Animal Management Association (Approval No. SYXK [Dian] 2014–0007) and the Experimental Animal Ethics Committee of the Institute (Approval No. DWSP201803014).The mice were maintained under a 12-h light/dark cycle (lights on at 08:00 h) at 22±5˚C. The animals were housed at five per cage and had free access to food and water. Prior to the experiments, the animals were routinely acclimated (>1 week) to the laboratory conditions to reduce potential stress effects during the experiments.

The experimental protocols for the rhesus macaques were approved by the Yunnan Provincial Experimental Animal Management Association (Approval No. SYXK [Dian] 2015–0006) and the Experimental Animal Ethics Committee of the institute (Approval No. DWSP201803013-2).The animals were reared in BSL-2 cages. The room was maintained with sufficient air, natural light and a temperature of approximately 22±5˚C. Food, water and fruits were supplied rationally. All efforts were to make the animals as comfortable as possible. All animals were confirmed to not be infected with herpes simian B virus or express any anti-HSV1 antibodies. The macaques were anesthetized using ketamine (10–20 mg/kg of body weight, Phoenix Pharmaceuticals, St. Joseph, MO, USA) intramuscularly prior to blood sampling and euthanasia.

### Cells

The African green monkey kidney Vero cell line (ATCC, Manassas, VA, USA), the human bronchial epithelial cell line 16HBE (ATCC), the human embryonic kidney cell line 293T (ATCC), the human glioma cell line Hs683 (ATCC), the human neuroblastoma cell line SH-SY5Y ATCC (and) the KMB17 cell line (IMB, CAMS, Yunnan, China) were maintained in high-glucose Dulbecco's modified Eagle's medium (DMEM; Corning, Corning, NY, USA) supplemented with 10% fetal bovine serum (HyClone, Logan, UT, USA). The culture medium was changed to DMEM supplemented with 2% fetal bovine serum after viral infection.

### Viruses

The HSV wild-type (WT) strain McKrae and the HSV1 mutant strains M3 [25] and M4 (*UL7-UL41-LAT-Us3*-MU) were used in the experiments. The mutants were verified by PCR and PCR product sequencing. Vero cells were used to detect viral titers. All virus-related experiments were performed under biosafety level (BSL) 2 conditions.

## Construction of recombinant mutant viruses

The method of recombinant mutant virus construction was similar to that used to establish the M3 strain and the protocol reported by Ran et al. [54]. The g-RNAs used were *Us12*-F1: CACCGGGTCTACTACGAGTCGGTG; *US12*-R1: AAACCACCGACTCGTAGTAGACCC; *US12*-F2: CACCCCAGGGATCCACGA CAACCC; and *US12*-R2: AAACGGGTTGTCGTG GATCCCTGG. The g-RNAs were annealed and inserted into the CRISPR/Cas9 system vector PX330 (Addgene, Cambridge, MA, USA). The plasmids were transfected into 293T cells, which were infected with the M4 strain. The newly established virus (M6) with partial *Us11 and Us12* deletion mutations was harvested from the infected 293T cells at 48 h post-infection (h.p.i.). The genomic regions surrounding the CRISPR target sites of the genes were PCR amplified with primers (*Us12*-sense, GGCAATGTGGAGATTCGG and *Us12*-antisense, CATGTGTCGGTGGGTTTG) using PrimeSTAR DNA polymerase (Takara, Dalian, Liaoning, China). The products were analyzed on 1.5% agarose gels and sequenced. After confirmation of the mutations, the mutated virus was cloned via a plaque assay. Moreover, specific primers surrounding and inside the mutated regions of the*Us3*, *Us11* and *Us12* genes were designed (S1 Table). The genomic regions surrounding or inside the deleted sites of the genes were PCR amplified with specific primers followed by agarose gel electrophoresis as above.

## Preliminary analysis of the M6virus

The HSV1 WT, M3 and M6 viruses were used to infect KMB17, 16HBE, SH-SY5Y and Hs683 cells at a multiplicity of infection (MOI) of 0.1 at 37˚C. The total viral yield from the cell culture supernatants and the infected cells was assessed at several time points (8, 16, 24, 32, 40 and 48 h.p.i.). The titers of all samples were determined by standard virus titration in Vero cells.

## Mouse study design

Four-week-old female BALB/c mice weighing 10–13 g (Vital River, Beijing, China) were purchased and housed in a BSL-2 facility of the Institute of Medical Biology. For an M6 virulence assay and RNA-sequencing analysis, mice were anesthetized under 2% inhaled isoflurane and infected with $10^4$ PFU of HSV1 WT strain McKrae,strain M6, strain M3 or PBS (sterile, pH 7.4; mock infection), which was used as a control, via the intranasal route. The mice were observed or weighed every 1 or 2 days. The nasal tissue and splenic lymphocytes were collected at particular time points for RNA-sequencing analysis. In addition, the nose, spleen and nervous system organs were collected at 1, 3, 7 and 30 days after viral infection, followed by viral load detection and pathological analysis.

Mice were anesthetized and infected via the intramuscular routes with three doses ($2x10^3$, $10^4$or $5x10^4$ PFU) of M6 strain or PBS (sterile, pH 7.4; mock infection), which was used as a control. At 7, 14 and 28 days post-infection, the blood was collected for peripheral blood lymphocytes separation and flow cytometry assay. At 14 and 28 days post-infection, the serum was collected for neutralizing antibody testing.

Mice were anesthetized and infected via the intramuscular routes with $10^4$ PFU of M3 strain, M6 strain or PBS (sterile, pH 7.4; mock infection), which was used as a control. At 25 days post-infection, the serum and splenic lymphocytes were collected for neutralizing antibody testing and ELISpot assays, respectively. Then, the infected mice were boosted as described above. At 25 days post-boost, the serum and splenic lymphocytes were collected. Moreover, all of the mice were challenged with the HSV1 WT strain McKrae ($1\times10^4$ PFU/ 50 µL/mouse) via the intranasal route at 30 days post-boost. Then, the mice were weighed every two days. The survival rate was assessed over a 10-day period. Tissue samples were

obtained at 3, 7 and 12 days after viral challenge and subjected to assessments of viral load and mouse organ pathology (Fig 4D).

## Preparation of mouse splenic lymphocytes

The assay was performed in accordance with standard protocols as described previously [55]. In brief, BALB/c mice were sacrificed after anesthetization with ether, the spleens were removed, and a single-cell suspension was prepared. Red blood cells were removed, and the isolated cells were washed and suspended in RPMI-1640 medium supplemented with 10% heat-inactivated fetal bovine serum, 100 U/mL penicillin and 100 μg/mL streptomycin.

## RNA-sequencing analysis

Mouse nasal tissues were removed at 1, 3 and 7 days after viral infection and stored in liquid nitrogen. Spleens were removed at 3, 7 and 30 days after viral infection and homogenized to obtain single-cell suspensions (Fig 3A). Total RNA was extracted using a TRIzol reagent kit (Invitrogen, Carlsbad, CA, USA) according to the manufacturer's protocol. RNA quality was assessed on an Agilent 2100 Bioanalyzer (Agilent Technologies, Palo Alto, CA, USA) and checked using RNase-free agarose gel electrophoresis. Then, the isolated mRNA was enriched with oligo(dT) beads by using the Ribo-Zero Magnetic Kit (Epicentre, Madison, WI, USA). The enriched mRNA was fragmented into short fragments using a fragmentation buffer and reverse transcribed into cDNA with random primers. Second-strand cDNA was synthesized with DNA polymerase I, RNase H, dNTP and a buffer. Then, the cDNA fragments were purified with a Qia Quick PCR extraction kit (Qiagen, Venlo, Netherlands), end repaired, treated for poly (A) addition and ligated to Illumina sequencing adaptors. The ligation products were separated by agarose gel electrophoresis, PCR amplified and sequenced using an IlluminaHiSeq 2500 by Gene Denovo Biotechnology Co. (Guangzhou, China). Then, the bioinformatic analysis included filtering of clean reads, alignment with ribosomal RNA (rRNA), alignment with a reference genome, transcript reconstruction, novel gene transcript identification and annotation, quantification of gene abundance, sample relationship analysis and differentially expressed gene (DEG) analysis. Genes with 2-fold or greater changes in expression at $p < 0.05$ in the Kyoto Encyclopedia of Genes and Genomes (KEGG) analyses were selected and grouped into functional categories. The KEGG pathways with $p < 0.05$, $p$-value in the lowest 20, related to the immune response are displayed in Table 1 and Table 2. To complement the KEGG analyses, genes with significant changes at each time point were also evaluated against the INTERFEROME database (http://www.interferome.org) to identify genes related to interferon (IFN) γ-mediated antiviral host responses [56]. The genes identified are shown in Fig 3B and 3C.

## Monkey experimental design and sample collection

Twenty-six approximately 1-year-old monkeys (IMBCAMS, Kunming, Yunnan, China) were used in the experiment. The experimental design and protocols were prepared as described previously [38]. The feeding and care of the animal were in accordance with standard protocols of our institute as described previously [38].

First, eight monkeys were infected via the intramuscular routes with three doses ($2x10^4$, $10^5$ or $5x10^5$ 50% cell culture infectious dose (CCID50)) of M6 strain or PBS (sterile, pH 7.4; mock infection), which was used as a control. At 7, 14 and 28 days post-infection, the blood was collected for peripheral blood lymphocyte separation and flow cytometry assay. At 14 and 28 days post-infection, the serum was collected for neutralizing antibody testing.

Then, the monkeys were immunized intramuscularly with $10^5$ CCID50 of M6 strain or PBS. At 2 months post-immunization, the monkeys were boosted as described above. Blood samples were collected from the immunized monkeys before the boost and at 1 month post-boost (Fig 6A). Two months post-boost, the immunized monkeys were challenged via the intranasal route with $10^6$ CCID50 of HSV1 WT strain McKrae. Blood, mouth swab and eye swab samples were collected from the infected monkeys every 3 days continuously for 12 days post-infection (Fig 6A). The swab samples were then centrifuged at 10,000 g for 10 min. The supernatants were used for qRT-PCR analysis to determine the viral load. Various tissues from the sacrificed animals were homogenized in PBS or a formalin solution (Solarbio, Beijing, China) and used for qRT-PCR analysis and pathological examination at 8 and 23 days after viral challenge and at 8 months after viral challenge.

## Flow cytometry analyses

Blood from the M6-immunized (three doses) mice or rhesus monkeys was collected at 7, 14 and 28 days post-infection. Red blood cells were removed, and the peripheral blood lymphocytes were washed and suspended in RPMI-1640 medium supplemented with 10% heat-inactivated fetal bovine serum, 100 U/mL penicillin and 100 μg/mL streptomycin. Then, the cells were cultured for 5 h in the presence of phorbol 12-myristate 13-acetate (PMA) (50 ng/ml; Solarbio), ionomycin calcium salt (1 μg/ml; Solarbio) and GolgiStop (0.67 μl/ml; BD Biosciences, New Jersey, USA). The cells were then treated with Fc block (mouse or human; BD Biosciences), followed by anti-CD3 (mouse or human; Biolegend, San Diego, CA, USA), anti-CD4 (mouse or human; Biolegend) and anti-CD8 (mouse; Biolegend, human; BD Pharmingen) and subsequently fixed and permeabilized using a fixation/permeabilization kit (BD Pharmingen) and stained with anti-IFN-γ (mouse; Biolegend, human; BD Pharmingen).

## Neutralization assay

A neutralization assay was performed in accordance with standard protocols as described previously [23]. In brief, serum samples were heat-inactivated, diluted and incubated with the WT virus for 2 h at 37˚C. The mixture was then incubated with Vero cells in 96-well plates at 37˚C. Cytopathic effects (CPEs) were observed after 1 week to determine the neutralizing antibody titer of each serum sample.

## IFN-γ-specific ELISpot assay

Splenic lymphocytes were isolated as described above [55]. A mouse or monkey IFN-γ ELISpot kit (MABTECH Inc., Cincinnati, OH, USA) was used according to the manufacturer's protocol. In brief, a plate was conditioned and seeded with splenic lymphocytes prior to the addition of 10 μg of stimulant (two peptides: gB498-505, SSIEFARL; and ICP6822-829, QTFDFGRL) (Sangon Biotech, Shanghai, China) [57–59]. The cells were then incubated at 37˚C for 30 h. Then, the cells were removed, and the plate was developed. The colored spots were counted.

## Histopathological examination

Mouse organs were fixed in 10% formalin and embedded in paraffin in tissue blocks. Approximately two slides per organ were stained with hematoxylin and eosin (H&E) to assess morphology.

## RT-PCR and quantitative (q) RT-PCR

The viral loads in the tissues were determined by qPCR with absolute quantification. Based on the methods of Ryncarz AJ et al. [60], the primers for the reaction bound within a region of the glycoprotein g gene in the HSV1 genome. The primers were *gG*-F, TCCTSGTTCCTMAC KGCCTCCC and *gG*-R,GCAGICAYACGTAACGCACGCT. In addition, a standard curve was produced from standard DNA samples (the p-GMT plasmid ligated to the *gG* gene fragment). Viral genomic DNA was extracted from mouse tissue using the Universal DNA Purification Kit (Tiangen). The TaqMan probe (Sangon Biotech, Shanghai, China) had the sequence 5′-6FAM-CGTCTGGACCAACCGCCACACAGGT-TAMRA. The reactions were performed using Premix Ex Taq (probe qPCR; Takara) on a Light Cycler system (Bio-Rad, Hercules, CA, USA).

To confirm the result of the RNA-sequencing analysis, the mRNA samples collected from infected mice at a specific time point used for the RNA-sequencing studies were analyzed individually by quantitative RT-PCR (qRT-PCR) to provide more detailed information about the genes of the relevant pathways. The expression levels of the genes were calculated by relative quantification using the comparative Ct method (ΔCt) with the mouse housekeeping gene GAPDH. Gene expression was expressed as the fold-change ($2^{-\Delta Ct}$) relative to the levels in samples from WT- or M6-injected mice, which were used for calibration. The total RNA from mouse tissues was extracted using TRIzol reagent (Life Technologies, Carlsbad, CA, USA), and amplification reactions were performed using a One-Step SYBR Prime Script™ PLUS RT-PCR Kit (Takara).Template-negative and RT-negative reactions served as controls, and the specific primer sets used are listed in (S2 Table).

## Statistical analysis

The experiments were performed in triplicate, and the results are expressed as the mean value with the standard error of the mean. SPSS software was used for statistical analyses. The weights of the infected mice were evaluated using a repeated measures analysis. A survival analysis was performed to analyze the survival rate of the infected mice. Differences in viral load between two groups were evaluated using an independent-samples *t*-test. A value of $p <$ 0.05 was considered significant.

## Supporting information

**S1 Text. The DNA sequence of the mutated *Us11* gene of the HSV1 M6 strain.** Supplement to Fig 1 to display the mutated sequencing details of the *Us11* gene of the M6 strain. (DOCX)

**S2 Text. The DNA sequence of the mutated *Us12* gene of the HSV1 M6 strain.** Supplement to Fig 1 to display the mutated sequencing details of the*Us12* gene of the M6 strain. (DOCX)

**S1 Table. Primers used for copy number determination of the *Us3*, *Us11* and *Us12* genes.** (XLSX)

**S2 Table. Primers used for q-RT-PCR of the confirmatory assay of RNA sequencing.** (XLSX)

**S1 Fig. Confirmation of *Us3*, *Us11* and *Us12* gene knockout in M6 mutant.** (A-C) Schematic for specific primer design of *Us3*(**A**), *Us11*(**B**) and *Us12*(**C**) genes. (**D-F**) PCR amplification of the M6 genome with specific primer pairs targeting the *Us3* (**D**), *Us11*(**E**) and *Us12*(**F**) genes

for verification of gene knockout.
(TIF)

**S2 Fig. Immune response-related pathway expression analysis in nose tissues of HSV1 WT- or M6-infected mice.** The nose tissues were collected at 1,3,7 days after viral infection followed by RNA extraction and q-RT-PCR assays. There was no difference at 1 d.p.i. **(A-C)** The gene expression of RNA samples at 3 d.p.i.**(D-F)**The gene expression of RNA samples at 7 d.p.i. Relative expression was measured by q-RT-PCR, and the column shows relative fold-change in expression compared with M6 (value = 1). mRNA levels were normalized using the geometric mean of GAPDH. The data are shown as the mean ± SEM based on data from three independent mice. $^*p < 0.05$; $^{**}p < 0.01$;$^{***}$ $p < 0.001$.
(TIF)

**S3 Fig. Immune response-related pathways expression analysis in splenic lymphocytes of HSV1 WT- or M6-infected mice.** The splenic lymphocytes were collected at 3,7 and 30 days after viral infection followed by RNA extraction and q-RT-PCR assays.**(A-C)**The gene expression of RNA samples at 3d.p.i.**(D,E)**The gene expression of RNA samples at 7 d.p.i. **(F,G)**The gene expression of RNA samples at 30 d.p.i. Relative expression was measured by q-RT-PCR, and the column shows the relative fold-change in expression compared with M6 or WT (value = 1). The mRNA levels were normalized using the geometric mean of GAPDH. The data are shown as the mean ± SEM based on data from three independent mice. $^*p < 0.05$; $^{**}p < 0.01$;$^{***}p < 0.001$.
(TIF)

## Author Contributions

**Formal analysis:** Xingli Xu, Xiao Feng, Lichun Zheng, Shengtao Fan, Ying Zhang, Dandan Li.

**Funding acquisition:** Xingli Xu, Min Feng, Ying Zhang, Qihan Li.

**Methodology:** Xingli Xu, Xiao Feng, Lichun Wang, Ting Yi, Guorun Jiang, Yun Liao.

**Project administration:** Xingli Xu, Qihan Li.

**Software:** Xingli Xu, Shengtao Fan, Ying Zhang.

**Supervision:** Xingli Xu, Min Feng, Qihan Li.

**Validation:** Xingli Xu, Qihan Li.

**Writing – original draft:** Xingli Xu.

**Writing – review & editing:** Min Feng, Qihan Li.

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
