## [Decision Letter · Decision Letter 0]

23 Feb 2020

Dear Dr. Li,

Thank you very much for submitting your manuscript "The assembly of mutations in 6 genes of HSV1 leads to an attenuated phenotype and induces immunity with a protective effect against viral infection in mice and rhesus macaques" for consideration at PLOS Pathogens. As with all papers reviewed by the journal, your manuscript was reviewed by members of the editorial board and by several independent reviewers. The reviewers appreciated the attention to an important topic. Based on the reviews, we are likely to accept this manuscript for publication, providing that you modify the manuscript according to the review recommendations.

Sincerely,

Pinghui Feng

Associate Editor

PLOS Pathogens

Blossom Damania

Section Editor

PLOS Pathogens

Kasturi Haldar

Editor-in-Chief

PLOS Pathogens

orcid.org/0000-0001-5065-158X

Michael Malim

Editor-in-Chief

PLOS Pathogens

orcid.org/0000-0002-7699-2064

Reviewer Comments (if any, and for reference):

Reviewer's Responses to Questions

**Part I - Summary**

Reviewer #1: The manuscript # PPATHOGENS-D-19-02372 entitled “The assembly of mutations in 6 genes of HSV1 leads to an attenuated phenotype and induces immunity with a protective effect against viral infection in mice and rhesus macaques” constructed an HSV-1 vaccine candidate M6 with mutations in the UL7, UL41, LAT, Us3, Us11, Us12 genes. The M6 HSV-1 replicates at lower rate and have a small plaque phenotype in culture cells, and an attenuated phenotype in mouse and rhesus macaques. Authors also state that when immunizing mice and monkeys with the mutant strain M6, it induces remarkable serum neutralizing antibody titers and T cells activation, and protect against HSV1 challenge by impeding viral replication, dissemination and pathogenesis.

The significant weekness of this study is that while M6 induces neutralizing antibodies, and reduce the viral replication, the level of neutralizing antibody titer is very low, and M6 does not prevent virus replication in the tissues, and latency establishment in sensory ganglia. The concludion drawn from this study is not in the accordence with the data presented.

Reviewer #2: In this article, the authors established M6 mutants with mutations in the UL7, UL41, LAT, Us3, Us11, Us12 genes as an experimental attenuated vaccine against HSV1. The mutant exhibited attenuated phenotype in an animal model. Furthermore, in mice and rhesus monkeys, the mutant enables to induce remarkable serum neutralizing antibody titers and T cell activation and protect against HSV1 challenge. However, there are some issues to be addressed before further consideration of this ms

Reviewer #3: In this study, the authors constructed an attenuated HSV1 virus strain with mutations in the UL7, UL41, LAT, Us3, Us11, Us12 genes. The animal models infected by the mutant virus, including mice and rhesus monkeys, had a remarkable T cell activation and neutralizing antibody. Moreover, the authors detected the proliferation characteristics and virulence phenotype in different cells and animal models. Overall, the study is important and interesting. However, some more experiments need to be done to strengthen their findings and conclusions.

**Part II – Major Issues: Key Experiments Required for Acceptance**

Reviewer #1: No new experiments are suggested, because additional experiment will not make this manuscript better.

Reviewer #2: 1. In addition to detecting neutral antibodies and responsive T cell proliferation in mice or rhesus monkeys, CD4+ and CD8+ T cell responses and of cytokines expression should also be detected by flow cytometry.

2. The authors should take M6 inoculation in mice and rhesus monkeys in a dose-dependent manner to examine indicators of specific immune responses.

3. To better reflect the reliability and authenticity of the research results, the authors should extend the experimental period and continuously observe the changes in neutralizing antibody levels and viral loads.

Reviewer #3: 1.In Figure 1, the authors constructed an HSV1 strain with the assembly of mutations in 6 genes. However, what are the copies of HSV1 genome in every single cell and how the authors make sure that the Us3, Us11 and Us12 genes knockout in all HSV1 genomes.

2.Compared with an HSV-1 mutant vaccine with a UL18 deletion (Acta Virol. 2018;62(2):164-71), what advantages and advances of the M6 mutant vaccine has in cells and animal models.

**Part III – Minor Issues: Editorial and Data Presentation Modifications**

Reviewer #1: The title of this manuscript suggests that M6 mutant induces immunity with a protective effect against viral infection in mice and rhesus macaques; however, the data clearly shows that it does not prevents viral infection in mice or rhesus.

Many sections of figures lack the titles or the details on the y axis.

In the introduction, authors stated that HSV-1 cause occasional genital herpes infection. Epidemiology indicates that more that 50% of first-time genital herpes infections are now caused by HSV-1 virus.

Reviewer #2: None

Reviewer #3: 1.In Figure 1C-G, the authors detected biological properties such as plaques number, growth curves of the M6 strain, M3 strain and WT strain. The authors could also observe the M3 infection of Balb/c mice to further strengthen their conclusions that a more significantly attenuated phenotype for the M6 strain than for the WT strain.

2.In Figure 3, the authors utilized omic analysis of mRNA expression to suggest a characteristic immune response initiated by M6. However, the conclusions seem unconvincingly unless the authors determine the gene expression by qPCR and ELISA.

3.In Figure 4, to confirm their hypothesis that mutation of some virally encoded proteins with functions inhibiting different key molecules in the innate and adaptive immune responses could lead to the host immune system developing an integrated response against viral antigens, the authors should apply flow cytometry technology to detect the T cell activation.

4.In Figure 4 and Table 3, the authors demonstrated that there is no difference between M3 - and M6-immunized mice, suggesting that M3 strain is enough for inducing specific neutralizing antibody and CTL responses. Therefore, compared with M3 strain, please clarify what are the advantages and improvements of M6 strain.

PLOS authors have the option to publish the peer review history of their article (what does this mean?). If published, this will include your full peer review and any attached files.

Reviewer #1: No

Reviewer #2: No

Reviewer #3: No
---

## [Decision Letter · Decision Letter 1]

13 Jun 2020

Dear Dr. Li,

We are pleased to inform you that your manuscript 'The assembly of mutations in 6 genes of HSV1 leads to an attenuated phenotype and induces immunity with a protective effect against viral infection in mice and rhesus macaques' has been provisionally accepted for publication in PLOS Pathogens. However, I would ask you to modify the title and conclusion according to the first reviewer (see comments below). Please let me know if you need any assistance on this.

Best regards,

Pinghui Feng

Associate Editor

PLOS Pathogens

Blossom Damania

Section Editor

PLOS Pathogens

Kasturi Haldar

Editor-in-Chief

PLOS Pathogens

orcid.org/0000-0001-5065-158X

Michael Malim

Editor-in-Chief

PLOS Pathogens

orcid.org/0000-0002-7699-2064

Reviewer Comments (if any, and for reference):

Reviewer's Responses to Questions

**Part I - Summary**

Reviewer #1: The revised manuscript # PPATHOGENS-D-19-02372R1 entitled “The assembly of mutations in 6 genes of HSV1 leads to an attenuated phenotype and induces immunity with a protective effect against viral infection in mice and rhesus macaques” designed an HSV-1 vaccine candidate M6 with mutations in the UL7, UL41, LAT, Us3, Us11, Us12 genes. Immunizing mice and monkeys with the mutant strain M6, neutralizing antibody titers and T cells activation, and protect against HSV1 disease.

Reviewer #2: The Authors have addressed all issues raised by this reviewer.

Reviewer #3: (No Response)

**Part II – Major Issues: Key Experiments Required for Acceptance**

Reviewer #1: The data presented in the study show M6 vaccine can prevent mice and rhesus from the HSV-1 disease (Figure 6C). Measurement of virus replication in eye swabs in HSV-1 challenged rhesus, both mock and vaccinated group showed similar level of viral copies (Figure 6F). The data show that vaccine can prevent disease but not the infection. As a matter of fact, the most vaccines against infectious diseases do not prevent infection, however they prevent disease. I propose the title and conclusion of this study should correctly reflect that. Title of this manuscript could be modified to “The assembly of mutations in 6 genes of HSV1 leads to an attenuated phenotype and induces protective immunity and prevent HSV-1 disease in mice and rhesus macaques.” The conclusion should also be revised accordingly.

Reviewer #2: The Authors have addressed all issues raised by this reviewer.

Reviewer #3: (No Response)

**Part III – Minor Issues: Editorial and Data Presentation Modifications**

Reviewer #1: (No Response)

Reviewer #2: None

Reviewer #3: (No Response)

PLOS authors have the option to publish the peer review history of their article (what does this mean?). If published, this will include your full peer review and any attached files.

Reviewer #1: No

Reviewer #2: None

Reviewer #3: No

---

## [Editor Report · Acceptance letter]

3 Aug 2020

Dear Dr. li,

We are delighted to inform you that your manuscript, "A HSV1 mutant leads to an attenuated phenotype and induces immunity with a protective effect ," has been formally accepted for publication in PLOS Pathogens.

Best regards,

Kasturi Haldar

Editor-in-Chief

PLOS Pathogens

orcid.org/0000-0001-5065-158X

Michael Malim

Editor-in-Chief

PLOS Pathogens

orcid.org/0000-0002-7699-2064